# Genome-Wide Identification and Characterization of *TCP* Gene Family Members in *Melastoma candidum*

**DOI:** 10.3390/molecules27249036

**Published:** 2022-12-18

**Authors:** Hui Li, Xiaoxia Wen, Xiong Huang, Mingke Wei, Hongpeng Chen, Yixun Yu, Seping Dai

**Affiliations:** 1Guangzhou Institute of Forestry and Landscape Architecture, Guangzhou 510405, China; 2College of Forestry and Landscape Architecture, South China Agricultural University, Guangzhou 510642, China; 3Faculty of Electronic Information Engineering, Guangdong Baiyun University, Guangzhou 510450, China; 4College of Forestry, Sichuan Agricultural University, Chengdu 611130, China; 5State Key Laboratory of Tree Genetics and Breeding, Chinese Academy of Forestry, Beijing 100091, China; 6School of Traditional Chinese Medicinal Resources, Guangdong Pharmaceutical University, Guangzhou 510006, China

**Keywords:** *TCP* gene family, phylogenetic analysis, collinearity analysis, gene pairs, *Melastoma candidum*

## Abstract

It has been confirmed that the plant-specific Teosinte-branched 1/Cycloidea/Proliferating (*TCP*) gene family plays a pivotal role during plant growth and development. *M. candidum* is a native ornamental species and has a wide range of pharmacodynamic effects. However, there is still a lack of research on *TCP*’s role in controlling *M*. *candidum*’s development, abiotic stress responses and hormone metabolism. A comprehensive description of the *TCP* gene family in *M*. *candidum* is urgently needed. In this study, we used the HMMER search method in conjunction with the BLASTp method to identify the members of the *TCP* gene family, and a total of 35 *TCP* genes were identified. A domain analysis further confirmed that all 35 TCPs contained a TCP superfamily, a characteristic involved in dimerization and DNA binding that can be found in most genes from this gene family, suggesting that our identification was effective. As a result of the domain conservation analysis, the 35 *TCP* genes could be classified into two classes, TCP-P and TCP-C, based on the conservative regions of 55 and 59 amino acids, respectively. Gene-duplication analysis revealed that most *TCP* genes were present in duplication events that eventually led to *TCP* gene expansion in *M. candidum*. All the detected gene pairs had a Ka/Ks value of less than one, suggesting that purification selection is the most important factor that influences the evolution of *TCP* genes. Phylogenetic analysis of three species displayed the evolutionary relationship of *TCP* genes across different species and further confirmed our results. The real-time quantitative PCR (qRT-PCR) results showed that *McTCP2a*, *McTCP7a*, *McTCP10*, *McTCP11*, *McTCP12a*, *McTCP13*, *McTCP16*, *McTCP17*, *McTCP18*, *McTCP20* and *McTCP21* may be involved in leaf development; *McTCP4a*, *McTCP1*, *McTCP14*, *McTCP17*, *McTCP18*, *McTCP20, McTCP22* and *McTCP24* may be involved in flower development; and *McTCP2a*, *McTCP3*, *McTCP5a*, *McTCP6*, *McTCP7a*, *McTCP9*, *McTCP11*, *McTCP14* and *McTCP16* may be involved in seed development. Our results dissect the *TCP* gene family across the genome of *M*. *candidum* and provide valuable information for exploring *TCP* genes to promote molecular breeding and property improvement of *M*. *candidum* in the future.

## 1. Introduction

TCP is a kind of plant-specific transcription factor (TF) with conserved sequences of 55 or 59 amino acids as a characteristic domain that is responsible for activating or repressing the transcription process and is involved in protein–protein interactions [1]. The *TCP* gene family was named after the TCP domain from the first identified members: TB1 (teosinte branched1 from maize) [2], CYC (cycloidea from *Antirrhinum*) [3], and PCFs (PCF proteins from rice) [4]. *TB1* could determine apical dominance in maize [2]. In *Antirrhinum majus*, *CYC* has been shown to play a role in floral bilateral symmetry [5]. The PCF1 and PCF2 proteins could bind to the promoter of *PCNA* to influence DNA replication and repair, etc. [4]. The TCP transcription factors (TFs) have been classified into two classes, namely Class I (also known as TCP-P) and Class II (also known as TCP-C), based on differences in their TCP domains [6]. TCP-P consists mainly of the PCF class, while TCP-C consists of two clades, namely the CIN clade and the CYC/TB1 clade [6,7]. Currently, it is thought that many of the identified TCP proteins harbor a ubiquitous TCP domain with a noncanonical basic helix–loop–helix (bHLH) structure, which is quite different from the DNA-binding basic helix–loop–helix structure [8]. A small number of TCPs also harbor an R domain, an 18–20 residue arginine-rich motif [7], acquired in most members of the CYC/TB1 family and barely in CIN members [9,10]. Several studies indicated that the R domain may form a hydrophilic α-helix [11].

TCP TFs are involved in a variety of biological processes, such as flower and leaf development [12,13], flower symmetry [14], shoot branching [15], leaf morphogenesis and senescence [16], circadian clock [17], stress response [18,19,20], etc., generally acting through plant-hormone-mediated signaling. In addition, TCP TFs could act downstream of hormonally mediated pathways as transcriptional modulators of the processes involved in cell division [21], or act upstream of plant hormones and influence levels of hormone synthesis, transport and signal transduction [22]. In *Arabidopsis*, AtTCP4 regulates jasmonic acid (JA) biosynthesis through interaction with the JGB to further mediate pollen germination and gametophyte development [23]. MPK8 interacts with AtTCP14 in the nucleus and phosphorylates AtTCP14 outside the nucleus to promote seed germination [24]. TCPs of Class II could form complexes with FT further and then bind on the promoter of *AP1* to regulate flowering [25]. Li et al. found that a group of AS2-binding TCP TFs, including TCP3, TCP4, TCP10 and TCP24, were regulators of leaf development by binding directly to the promoter of *BP* and *KNAT2* to repress their expression [26]. *TCP* genes can also improve plant resistance and adaptation. For example, AtTCP20 could activate nitrate-assimilatory-related genes to improve their expression level under nitrogen starvation conditions [18]. Overexpression of *DgTCP1* improved the cold tolerance of chrysanthemum when comparing wild types [19]. In *Oryza sativa*, OsPCF2 could activate the expression of *OsNHX1* to promote salt and drought tolerance [20]. The expression of some *TCP* genes is regulated by miR319 to influence leaf shapes and petal development. In the *JAGGED AND WAVY* (*JAW*-D) mutant, for instance, overexpression of *miR319a* leads to low expression of five class II members, *TCP2*, *TCP3*, *TCP4*, *TCP10* and *TCP24*. The transgenic plants exhibited phenotypic defects, including highly serrated leaves, petal development changes and delayed leaf senescence [13,16,27].

In some species, *TCP* genes have been identified, and they vary greatly from species to species. For example, 24 *TCP* gene members were found in *Arabidopsis* [6]; 30 *TCP* member genes were found in *Solanum lycopersicum* [28]; 27 *TCP* genes were identified in *Citrullus lanatus* [29]; 23 *TCP* genes were found in *Halaenopsis equestris* [30]; 39 *TCP* genes had been confirmed in *Brassica rapa* ssp [31]; 75 *TCP* genes were found in *Gossypium barbadense* [32]; 21 *TCP* genes were found in *O. sativa* [33]; 27 *TCP* genes were found in *Cucumis sativu* [34]; 30 *TCP* genes were found in tomato [28]; 46 *TCP* genes were found in *Zea mays* L. [35]; 20 *TCP* genes were confirmed in peach [36]; 19 *TCP* genes were found in *Fragaria vesca* [37]; 33 *TCPs* were identified in *Populus euphratica* [38]; and 17 *TCP* genes were confirmed in *Vitis vinifera* [39]. Although *TCP* genes have important regulatory roles in plant growth and development, abiotic stress responses and hormone metabolism, limited information is available on *M*. *candidum.* Here, we performed a comprehensive bioinformatic analysis of the *TCP* gene family based on the genome of *M*. *candidum.* Characterization of *TCPs* in *M*. *candidum* was carried out based on motif, domains and gene structure analyses. Key conserved domains in each type of TCP protein were confirmed. Duplication events of *TCP* genes were investigated through collinearity analysis within the genome. Synonymous substitution (Ks) and nonsynonymous substitution rates (Ka) and their ratio (Ka/Ks) were calculated to confirm the divergence time and major force to promote the evolution of *TCP* genes pairs in *M*. *candidum*. Evolutionary analyses among *Arabidopsis*, *Populus* and *M*. *candidum* were performed to illustrate the gene relationships of the three species. qRT-PCR was employed to investigate the expression pattern of the selected *TCP* genes in different tissues and developmental stages. Our research will assist in better comprehending the classification and expression pattern of *TCP* genes in *M*. *candidum* and provide valuable information for studying the functions of the *TCP* gene family during development or abiotic stress to further use them in the molecular breeding of *M*. *candidum*.

## 2. Results

### 2.1. Chromosome Distribution and Evolution Relationship of TCP Genes

We first conducted an HMMER search through the genome of *M. candidum* and a total of 35 *TCP* gene family members were obtained. As a result of the distribution analysis of the 35 identified *TCP* gene members, it was found that all 35 members were unevenly distributed within 11 out of 12 chromosomes, with the exception of Chr11. With seven members and five members, Chr07 and Chr12 were the main TCP carriers with the highest proportion of *TCP* gene distribution, respectively occupying 20% and 14.3% of the total chromosomes. There were fewer *TCP* genes present in Chr02, Chr03, Chr04 and Chr06, as compared with other chromosomes; only two *TCP* genes were distributed (Figure 1). A phylogenetic tree for these 35 members of the *TCP* gene family was constructed by MEGA6.01 based on the amino acid sequences extracted from the genome file. The phylogenetic result showed that the 35 *TCP* gene family members could be divided into three main clades: TCP-P (Class I), CIN and CYC/TB1 (Figure 2). There were 11 subfamily members found in TCP-P, including *McTCP6*, *McTCP7*, *McTCP8*, *McTCP9*, *McTCP11*, *McTCP14, McTCP16*, *McTCP19*, *McTCP20*, *McTCP21* and *McTCP22*. Within the TCP-P group, there were also three sub-clades that can be distinguished between the members. Amongst, *McTCP9* was found to be divided into a single clade, suggesting that it differs greatly from other *TCPs*. In comparison to other *TCP* genes, *McTCP19*, *McTCP20*, *McTCP7*, *McTCP22* and *McTCP14* were close to each other. CYC/TB1 and CIN are both members of the TCP-C (Class II) group. Three main members of the *TCP* gene family had been identified in the subgroup CYC/TB1, including *McTCP1*, *McTPC12* and *McTCP18*. Moreover, in our detailed divisions, we determined that *McTCP1*, *McTCP12b* and *McTCP12e* were classified as one group, while *McTCP12a*, *McTCP12c*, *McTCP12d* and *McTCP18* were classified as another group. Among the members of the CIN subgroup, eight members of the *TCP* gene family had been identified, including *McTCP2*, *McTCP3*, *McTCP4*, *McTCP5*, *McTCP10*, *McTCP13*, *McTCP17* and *McTCP24*. Amongst them, approximately three sub-clades could be divided. *McTCP5*, *McTCP13* and *McTCP17* were classified into one clade; *McTCP2* and *McTCP24* were grouped into one group; *McTCP3*, *McTCP4* and *McTCP10* were classified into a single clade (Figure 2).

### 2.2. Motif, Domain, Gene Structure and Promoter Analysis of TCP Genes in M. candidum

To further verify the identified 35 *TCP* gene members, we performed motif, domain and gene structure analysis. In the promoter region, genes in four clades (C1–C4) showed obvious motif characteristics: adjacent motif 1 and motif 2; genes in clade five and clade six displayed obvious adjacent motifs of 1 and 3. According to domain analysis, the most common characteristics of *TCP* gene family members were TCP superfamily proteins, which was a key characteristic that distinguishes the *TCP* gene family from other gene families. This indicates that all the 35 members belonged to the *TCP* gene family. There was, however, a great deal of variation in the TCP superfamily proteins between the different types of TCP individuals. The results of gene structure analysis showed that all the *TCP* genes in *M. candidum* contained only one exon, and more than 50% of *TCP* genes lack UTR annotation, which might be caused by the assembling quality of the genome (Figure 3).

The promoter analysis results showed that the promoter region of some *TCP* genes contained plant hormone-response elements. For example, 60% (21 out of 35) of *TCP* genes promoters contained a TGA element, which is an auxin-responsive element; 62.86% (22 out of 35) carried a TCA element, which is a salicylic-acid-responsiveness element; 60% (21 out of 35) contained an O2 site element, which is a zein-metabolism-regulation element; 91.43% (32 out of 35) harbored an ABRE-binding site, which is an abscisic-acid-responsiveness element; 80% (28 out of 35) harbored an MeJA motif, which is an MeJA-responsiveness element, etc. Promoters of some genes contained a stress-responsive element. For instance, 65.71% (23 out of 35) of *TCP* gene promoters carried LTR, a low-temperature-responsiveness element; 54.29% (19 out of 35) had an MBS-binding site, a drought-inducibility binding site; 48.57% (17 out of 35) contained a TC-rich repeat, a defense stress-responsiveness element; only 5.71% (2 out of 35) contained a WUN motif, a wound-responsive element (Figure 4 and Figure 5). All sequences of the bind sites are listed in Appendix A. We also observed that the promoter of *TCP2b* included an MBSI element, which controls flavonoid biosynthesis (Appendix A).

### 2.3. Conserved Region Analyses of the Identified TCP Proteins

We aligned the protein sequences of 35 TCPs to find their conserved region by the ClustalW method. The results showed that these 35 TCPs were not conserved well enough (Figure 6), indicating that there are functional differences between members of the *TCP* gene family (Figure 3). This was consistent with the results of the domain analysis of TCPs in other species. An earlier study has shown that TCP TFs can be classified into two classes, Class I and Class II [6]. In both of the classes, the N-terminus of the proteins is characterized by a basic-helix–loop–helix structure motif [40]. According to our aligned sequences of these genes, we found that they could be roughly divided into two classes. The first group of TCPs contained McTCP6*,* McTCP7a, McTCP7b, McTCP8, McTCP9, McTCP11, McTCP14, McTCP16, McTCP19a-c, McTCP20, McTCP21 and McTCP22 (Appendix A). These 14 TCPs are highly conservative and belong to the TCP-P type with a 55 aa-long conserved region. The second group included McTCP1, McTCP2a, b, McTCP3, McTCP4a–d, McTCP5a–c, McTCP10, McTCP12a–e, McTCP13, McTCP17, McTCP18 and McTCP24. These 21 TCPs are highly conservative at the 7 aa to 24 aa positions (Appendix A) and belong to the TCP-C type with a 59 aa-long conserved region. In both groups, their conservation parts were comprised of Basic Helix I–Loop–Helix II structures. Our conservation analysis results illustrated that these TCPs may have evolved into two functionally different groups in *M. candidum*. In addition, we observed that seven TCPs in CYC/TB1 also contained another conserved domain, namely the R domain, with six of these genes having an R domain with a length of 18 aa, whereas McTCP18 only had a 14 aa-length R domain (Figure 6b).

### 2.4. Duplication Events and Divergence Time Estimation of TCP Gene Pairs

To investigate the duplication events of identified *TCP* genes, we conducted a collinearity analysis of these *TCP* genes from the genome level by using TBtools software. Figure 7a shows the associated gene pairs of *TCP* genes. A total of 34 out of 35 genes were found to have corresponding genes, indicating that *TCP* genes have gone through extensive duplication events. To understand the divergence time of gene pairs, we calculated the synonymous substitution rate (Ks/dS) and estimated the divergence time by using a divergence rate of 6.5 × 10^−9^ per synonymous site per year [41]. There was a wide range of divergence times between 12.8 and 99.82 million years ago for the *TCP* genes (Figure 7b). The earliest duplication event of *TCP* genes in *M. candidum* was *McTCP18* and *McTCP12d*, which happened with a divergence of about 99.82 MYA. Most of the members of the same sub-family have undergone duplication events in recent years. For example, *McTCP24* and *McTCP2a* diverged at about 12.8 MYA; *McTCP4b* and *McTCP3* diverged at about 22.01 MYA; *McTCP7a* and *McTCP21* experienced duplication events at about 21.26 MYA; and *McTCP20* and *McTCP16* diverged from each other at 18.21 MYA. There were, however, some members of the same sub-families that separated a long time ago as well. For example, *McTCP4a* and *McTCP3* diverged at about 81.85 MYA; *McTCP12b* and *McTCP12e* separated at 74.58 MYA; and *McTCP4c* and *McTCP4d* broke away at 82.83 MYA. Combining phylogenetic trees, we found that the longer the divergence time, the more distant evolutionary relationships were. A substitution ratio mutation (Ka/Ks) reflects the selection method experienced by gene pairs. When Ka/Ks < 1, the genes experience purifying selection, which means the selection process could neutralize mutation to maintain the stability of the protein; in contrast, when Ka/Ks > 1, the genes experience positive selection, which means great mutation happens in genes and eventually leads to a change in coded proteins. Our identified *TCP* gene pairs had Ka/Ks values ranging from 0.1 to 0.33, proving that all of these genes experienced a purification selection process in *M. candidum* (Figure 7b). This reflected that *M. candidum* experienced little severe mutation disturbance during its life cycle on earth.

### 2.5. Evolutionary Relationship of TCP Genes among Different Species

Eukaryotic genomes differ in the degree to which genes remain on corresponding chromosomes (synteny) and in corresponding orders (collinearity) [42]. Comparative analysis of species genomes could illustrate genomic evolution. Species relationships could be studied by searching for conserved genes pairwise among them [7]. To understand the evolutionary relationships of *TCP* genes among different species, we conducted a multi-collinearity analysis by selecting herbal species *A. thaliana* and woody species *P. trichocarpa*. A total of 71.4% (25 out of 35) *TCP* genes of *M. candidum* had a collinearity connection with 18 *TCP* genes of *A. thaliana*, and 74.3% (26 out of 35) *TCP* genes of *M. candidum* had a collinearity connection with 20 *TCP* genes of *P. trichocarpa* (Figure 8). According to the phylogenetic tree, these TCP genes were divided into three main groups. The first clade was mainly comprised of TCP-P group members, such as *TCP9* (At2g45680, Potri.003G120201, Potri.001G111800, *McTCP9*), *TCP19* (At5g51910, *McTCP19*, Potri.012G135900), *TCP6* (At5g41030, *McTCP6*), *TCP20* (At3g27010, Potri.001G327100, Potri.003G167900, Potri.001G060000, *McTCP20*), *TCP14* (At3g47620, *McTCP14*), *TCP15* (At1g69690, *McTCP15*), *TCP22* (At1g72010, Potri.019G081800, *McTCP22*) and *TCP23* (At1g35560, Potri.013G110700) (Figure 9). It was interesting to note that *McTCP9* is highly homologous to At2g45680 (*AtTCP9*), whereas it was closely clustered with *TCP19* of *Arabidopsis* and *P. trichocarpa* in the phylogenetic tree*. McTCP19* was highly homologous to AT5G51910 (*AtTCP19*), but it was closely clustered with *TCP9* of *Arabidopsis* and *P. trichocarpa*. This evidence showed that *TCP9* and *TCP19* had relatively close evolutionary relationships. The second clade mainly comprised CYC/TB1-type *TCP* genes, such as *TCP1* (Potri.017G112000), *TCP12* (Potri.015G050500, Potri.012G059900, Potri.008G115800, Potri.010G130200, At1g68800, *McTCP12b*–*e*) and *TCP18* (At3g18550, *McTCP18*). Interestingly, in this clade, we only obtained one *TCP1* gene in *P. trichocarpa,* indicating a low collinearity relationship for this gene among three species and high similarity of *TCP1* in *P. trichocarpa* with other *TCP* genes such as *McTCP12e* in the other two species (Figure 9). More *TCP12* genes were found in *Populus* and *M. candidum*, suggesting that more complex gene-duplication events for *TCP12* happened in these two species than in *Arabidopsis*. The third clade mainly comprised CIN-type *TCP* genes, such as *TCP4* (Potri.011G096600, Potri.019G091300, At3g15030, *McTCP4a*–*d*), *TCP3* (Potri.001G375800, At1g53230, *McTCP3*), *TCP13* (Potri.017G094800, At3g0215, *McTCP13*), *TCP5* (Potri.015G058800, At5g60970, *McTCP5a*–*c*), *TCP10* (At2g31070, *McTCP10*), *TCP2* (At4g18390, *McTCP2a* and *b*) and *TCP24* (At1g30210, *McTCP24*) (Figure 9). Amongst them, *McTCP4* and *McTCP5* had more members, suggesting that these two subfamilies experienced more complex gene-duplication events in *M. candidum* than in *Arabidopsis* and *P. trichocarpa*. *TCP2*, *TCP4*, *TCP10*, *TCP14*, *TCP15* and *TCP18* were only found in collinearity relations between *M. candidum* and *Arabidopsis,* reflecting that these genes were conserved in these two species. In each clade, we observed a closer evolutionary relationship intraspecies than interspecies. In all, our results still proved the similar characteristics of *TCP* genes in different species, and the *TCP* genes identified in *M. candidum* were reliable.

### 2.6. Expression Patterns of the Identified TCP Genes in Different Tissues of M. candidum

To understand the expression pattern of *TCP* genes, we took part of them to perform qRT-PCR experiments with cDNA as templated from nine kinds of tissues of *M. candidum*. These nine tissues included young leaves (YL), adult leaves (AL), young stems (YS), adult stems (AS), seeds (S), roots (R), early-stage flowers (EF), middle-stage flowers (MF) and blooming flowers (BF) (Figure 9). Interestingly, our study suggested that more than half of the selected TCP genes, including *McTCP2a*, *McTCP7a*, *McTCP10*, *McTCP11*, *McTCP12a*, *McTCP13*, *McTCP16*, *McTCP17*, *McTCP18*, *McTCP20* and *McTCP21*, were highly expressed in adult leaves, indicating that they could play important roles in leaf development. Some *TCP* genes had low expression levels in all nine tissues, such as *McTCP1*, *McTCP5a*, *McTCP6*, *McTCP9* and *McTCP19a*. Even though *McTCP5a*, *McTCP6* and *McTCP9* exhibited low levels of expression in all the tissues tested, they still displayed higher expression levels in seeds than in the rest tissues. Several *TCP* genes, including *McTCP2a*, *McTCP3*, *McTCP5a*, *McTCP6*, *McTCP7a*, *McTCP9*, *McTCP11*, *McTCP14* and *McTCP16,* had high expression levels in seeds. It was found that all of the selected *TCP* genes had low expression levels in roots and young leaves. In the early stages of flower development, *McTCP4a*, *McTCP10*, *McTCP17*, *McTCP18* and *McTCP24* were highly expressed; in the middle-stage flowers, *McTCP14* was highly expressed; and in the blooming flowers, *McTCP20* and *McTCP22* were highly expressed (Figure 10). These results reflected the different roles of *TCP* gene family members during flower development.

## 3. Discussion

TCP TFs play important roles in diverse biological processes [43,44], making them promising candidates in molecular breeding. Although the *TCP* genes have been identified in some model species, such as *Arabidopsis*, rice, maize, *Populus,* etc., the identification of *TCP* genes in *M. candidum* has still not been reported because of the absence of complete genome resources. In this study, we integrated the HMMER search and BLASTp method to identify *TCP* gene family members based on the genome of *M. candidum*. Combining domain analyses, a total of 35 TCP TFs were identified and were classified into two typical classes of TCPs, TCP-P and TCP-C. Most *TCP* genes were found to have been duplicated, resulting in their expansion in *M*. *candidum*. The expansion of *TCP* genes may reflect the involvement of these genes in a more complicated transcription process in the perennial shrub species.

The analysis of the phylogenetic trees of *TCPs*—in *M. candidum* and multi-species—supports the results of earlier related research [2,3,4], which has declared two subfamilies are included in *TCP* gene family. In addition, we have also detected conserved regions in our identified TCP proteins that are similar to those reported previously, which illustrates that there is a high degree of conservation of TCPs across species. However, there were also some subtle differences. For example, several studies have suggested that the R domain of the R proteins is primarily responsible for mediating the interactions of proteins [45]. In contrast, in CIN- and TCP-P-type TFs, we did not yet find the R domain; this is different from TCP identification in bananas [7,45], indicating different characteristics of TCPs between *M. candidum* and bananas. It is important to note, however, that there are different numbers or types of TCPs in different species. A number of previous studies have also revealed that there are a large number of differences between different species, ranging from 17 to 75 family members [6,31,32,35,37,39]. During the course of our study, a total of 35 *TCP* gene members were identified. However, comparing 24 *TCP* genes in *Arabidopsis*, we observed that *TCP15* and *TCP23* were not present in our results. When we used 24 *TCP* genes of *Arabidopsis* as a query to blast against the database of the *M. candidum* genome, we found that all 24 *TCP* genes of *Arabidopsis* were able to find the best match in *M. candidum* (Appendix A). Furthermore, all of the best matches in this study belonged to the *TCP* genes we had identified in *M*. *candidum*. Some *TCP* genes in *Arabidopsis*, such as AT1G53230 (*AtTCP3*) and AT3G15030 (*AtTCP4*), were best matched with *McTCP3*, and AT2G37000 (*AtTCP11*), AT3G47620 (*AtTCP14*) and AT1G69690 (*AtTCP15*) were best matched with *McTCP11,* illustrating that gene-duplication events also happened in *Arabidopsis*. Hence, it is normal that we might not be able to find all *TCP* genes in our results because of the different rules for naming genes. In regards to the different number of genes identified in a specific family, it has been suggested that it may be due to the differences in the species themselves; for example, gene-duplication events and the size of the genome in one species could have an effect on the number of genes identified in a specific family [45]. The second reason is because of the threshold that was chosen when performing the HMMER search. There are some researchers who broaden the definition of the *E* value from 0.01 to 0.10 [10]. There is no doubt that the results will be different if the threshold is set at a different level. As of now, researchers only name genes numerically, which results in the loss of gene structure information, as well as the existence of gene duplications. Therefore, in a related field, authoritative and reasonable naming standards are urgently needed.

The TCP transcription factors are ancient proteins that are specific to plants. Although there have been no reports in unicellular algae, they have been reported in pluricellular green algae, moss, ferns and lycophytes, typically with five to six members in them [10,43]. Gene family expansion and evolution are mostly attributed to gene-duplication events such as segmental, tandem, transposition and whole-genome-duplication (WGD) events [46,47]. Segmentally duplicated genes are those that are present on different chromosomes and show similar expression patterns [48]. According to this concept, we found that many gene pairs were also distributed across different chromosomes as well, such as *McTCP8*/*McTCP12*, *McTCP17*/*McTCP5a* and *5b*, *McTCP3*/*McTCP4a* and *4b*, etc., indicating that segmental duplication events play important roles in the *TCP* gene member expansion process of *M. candidum* (Figure 7a). The results of this study are in agreement with previous investigations of the *TCP* gene family in other species [35,38,49]. In most plants, WGD is also the dominant cause of genome diversity [7]. Based on synonymous substitution rate (Ks) analysis, previous studies had defined different types of WGD events such as α + β WGD and γ WGD events [45,50]. Using this method, we were able to determine that the genome of *M*. *candidum* has undergone three WGD events during evolution. Amongst them, approximately 41.7% (10 out of 24) *TCP* gene pairs experienced WGD_α_ events (Ks < 0.45), 16.7% (4 out of 24) of *TCP* gene pairs experienced WGD_β_ events (0.45 ≤ Ks ≤ 0.85) and 41.7% (10 out of 24) *TCP* gene pairs experienced WGD_γ_ events (Ks > 0.85) (Figure 7b), suggesting WGD is responsible for the expansion of the *TCP* gene family in *M. candidum*. A gene pair with a Ks greater than 0.85 indicates that the gene may have originated from a more ancient duplication event and has since undergone multiple rounds of WGD. Earlier studies have shown that *TCP* gene expansion is not uniformly distributed in various classes of bananas. For example, the CIN subclade exhibited more gene duplication than any other subclade [7]. However, in our results, we did not observe uneven distribution of gene-duplication events in three types of *TCP* genes. There were duplication events detected in most of the *TCP* genes of different types (Figure 7a,b). It is a possibility that this may be caused by the species itself.

There is a relationship between the expression profile of genes and the function of those genes [51]. The genes had distinct expression profiles in diverse organs, suggesting their role in the development of various organs. There is a great deal of evidence that TCP transcription factors play an important role in plant growth and development, including the development of all types of branches, leaves and flowers [21,45], as well as fruit development and ripening [37]. Some research had proved that more than two thirds of *TCP-C* subfamily genes have organ-specific expressions [10] and could inhibit plant growth and cell differentiation [22]. *TCP-P*—and some *CIN*—genes were detected in the flower, leaf and stem of *Prunus mume* [10]. In *Arabidopsis*, eight CIN-type genes, including *AtTCP2*, *AtTCP3*, *AtTCP4*, *AtTCP5*, *AtTCP10*, *AtTCP13*, *AtTCP17* and *AtTCP24,* are highly transcribed in the leaf and are responsible for the regulation of leaf growth [16,52]. *CYC* genes may be related to changes in petal size [53] and floral zygomorphy [54]. In accordance with the previous study, our results of qRT-PCR assays revealed that some members of the TCP-P sub-family—such as *McTCP6*, *McTCP8*, *McTCP14* and *McTCP22*, —and some members of the CYC sub-family—such as *McTCP1* and *McTCP8*—were highly expressed in the stems of *M*. *candidum*. Several TCP-P members—such as *McTCP7a*, *McTCP8*, *McTCP9*, *McTCP11*, *McTCP14*, *McTCP16*, *McTCP20* and *McTCP21*—CYC members—such as *McTCP1*, *McTCP12a* and *McTCP18*—as well as *TCP-C* gene family members—such as *McTCP2a*, *McTCP3*, *McTCP5a*, *McTCP10*, *McTCP13* and *McTCP17*—were highly expressed in leaves of *M*. *candidum*. There was a high-level expression of CIN sub-family members such as *McTCP3*, *McTCP4a*, *McTCP8*, *McTCP10* and *McTCP20* in the flowers of *M*. *candidum* (Figure 9). These results indicate that *TCP* genes may be functionally conserved among different species.

## 4. Materials and Methods

### 4.1. Identification of TCP Transcription Factors

The genome resource of *M. candidum* was provided by Zhou’s group of Sun Yat-sen University (unpublished data). To identify *TCP* gene family members in *M*. *candidum*, the TCP domain HMM profile (PF03634) was used as a query to perform a HMMER search with an *E*-value cut-off of 1 × 10^−3^ through the *M*. *candidum* genome by following the HMMER User Guide. All the motif analyses for the obtained proteins were conducted on MEME Suite Version 5.5.0. Available online: https://meme-suite.org/meme/tools/meme (accessed on 24 August 2022) and domain analyses were performed on the PFAM website (https://www.ebi.ac.uk/Tools/pfa/pfamscan/) with an *E*-value 1 × 10^−5^ to further verify the identified *TCP* gene members.

### 4.2. Phylogeny Tree Construction and Location of TCP Gene Family Members in Chromosome

All the identified *TCP* genes were aligned by MUSCLE method in MEGA 6.01, and then a phylogenetic tree was constructed by the ML (maximum likelihood) method based on LG models with 1000 bootstrap replications. The phylogeny tree was visualized on the Interactive tree of life (iTOL). Available online: https://itol.embl.de/ (accessed on 26 August 2022). By utilizing the Gene Location Visualize from the GTF/GFF function module of TBtools software, we visualized the distribution of *TCP* genes along the chromosomes through the gtf annotation of the genome and the gene density file.

### 4.3. Visualization of Motif, Domain, Gene Structure and Promoter of TCP Genes

Motif, domain, gene structure and promoter analyses were conducted for all the identified *TCP* genes. The upstream 2000 bp of the *TCP* CDs were extracted for the purpose of conducting promoter analyses on PlantCARE. Available online: https://bioinformatics.psb.ugent.be/webtools/plantcare/html/ (accessed on 28 August 2022). With the .gff file of the genome, gene structures including CDs and untranslational region (UTR) were displayed in the gene structure view of TBtools. Based on the identification of TCP transcription factor parts, the motifs and domains of *TCP* genes were visualized by the gene structure view and Batch SMART module of the TBtools software, respectively.

### 4.4. Conserved Region Analysis of the TCP Proteins

The protein sequences of identified TCPs were first aligned by the ClustalW method in MEGA software. The aligned fasta file was input into Jalview software to show the conserved region of all the aligned TCPs. The Seqlogo graph of the conserved region was produced by the Amazing Simple SeqLogo module of TBtools to show the conservation of amino acids in the corresponding region.

### 4.5. Identification of TCP Gene Pairs and Divergence Time Estimation

Gene collinearity analysis and visualization within the *M*. *candidum* genome were conducted by TBtools [55]. In a nutshell, (1) the collinearity, CTL, and .gff file were produced by the One Step MCScanX-Super Fast module; (2) the chromosome length file was generated by the Fasta stats module; (3) a gene pair file was produced from the collinearity file by the File Merge for MCScanX module; (4) a link region file was generated by the File Transformat for Microsyteny viewer module; (5) gene pairs of *TCP* gene family members were produced by the Text Block Extract and Filter module. Based on the above analyses, we aligned protein sequences and ORFs of the *TCP* gene pairs by ClustalW method in MEGA software. The synonymous substitution (Ks) and non-synonymous substitution (Ka) rates were calculated using the CODEML program of PAML on PAL2NAL. Available online: http://www.bork.embl.de/pal2nal/ (accessed on 29 August 2022) [56]. Divergence times (DT) of the gene pairs were estimated using the formula T = Ks/2λ, with the divergence rate λ = 6.5 × 10^−9^ [41,57]. *TCP* gene pairs were visualized in the Advanced circos module of the TBtools [55].

### 4.6. Collinearity and Phylogeny Analyses of TCP Genes among Different Species

Multiple Chr layout, gene link and .gff files between *A. thaliana* and *M*. *candidum, P. trichocarpa* and *M*. *candidum* were produced by the One Step MCScanX-Super Fast module of the TBtools with an *E*-value of 1 × 10^−3^. The homologous genes among the different species were obtained from the merged gene link file after merging three files of two comparison groups using the File Merge For MCScan-X module. By using the extracted protein sequence of these homologous genes, a phylogeny tree among three species was performed in MEGA by the ML method with 1000 boot replications. *TCP* gene collinearity plots among different species were visualized by the Multiple Synteny Plot module of the TBtools [55].

### 4.7. Quantitative Real-Time PCR and Statistical Analysis of the Selected TCP Genes in Different Tissues of M. candidum

A total of 22 selected *TCP* genes (Appendix A) were blasted against the whole genome of *M. candidum* to find a specific region, and then the primers were designed for qRT-PCR. Nine tissues, including young leaves (YL), adult leaves (AL), young stems (YS), adult stems (AS), seeds (S), roots (R) and three-stage flowers (EF, MF, LF), were collected to extract the total RNA by OminiPlant RNA Kit (DNase I) (CW2598, CWBIO, Taizhou, China) following the manufacturer’s instructions. After checking the quality of the total RNAs, 0.5 μg of total RNA was reverse-transcribed into first-strand cDNA using the PrimeScript RT reagent Kit gDNA Eraser (Takara, Dalian, China). Then, the SYBR @Premix Ex Taq TMII (Takara, Dalian, China) was used for qRT-PCR of genes, following the manufacturer’s instruction, on the Illumina Eco real-time PCR system (Illumina, USA) platform. The *α-tubulin* gene of *M. candidum* was used as the internal reference gene. Ct values obtained on the thermal cycler platform were then calculated by the 2^−ΔCt^ algorithm [58]. Using mixtures of cDNAs from nine tissues, primer PCR amplification efficiency was evaluated by following Pfaffl’s research [59]. Primers sequences, length of PCR products and PCR amplification efficiency for each pair of primers are listed in Appendix A. Relative expression levels of selected genes in different tissues were analyzed using ANOVA with Duncan’s test by SPSS 18.0 software. The graphs were visualized by GraphPad Prism 9 and Adobe Illustrator 2020.

## 5. Conclusions

Collectively, a total of 35 *TCP* gene family members were identified based on the genome-wide identification of *M. candidum* in our study. In the identified *TCP* gene family members, there was a common domain in the TCP superfamily. Despite the fact that there were differences between all the *TCP* genes, the same gene type showed high conservation. The number of *TCP* genes was more due to more frequent gene-duplication events that occurred in *M*. *candidum*. It is more likely that the majority of *TCP* genes had been affected by natural selection rather than human interference. *TCP* genes showed distinct family members from two model species. A greater focus should be placed on functional exploration to expand their application in the garden and pharmaceutical industries. Our results provided valuable information for understanding the classification and functions of *TCP* genes in *M. candidum*.

## Figures and Tables

**Figure 1 molecules-27-09036-f001:**
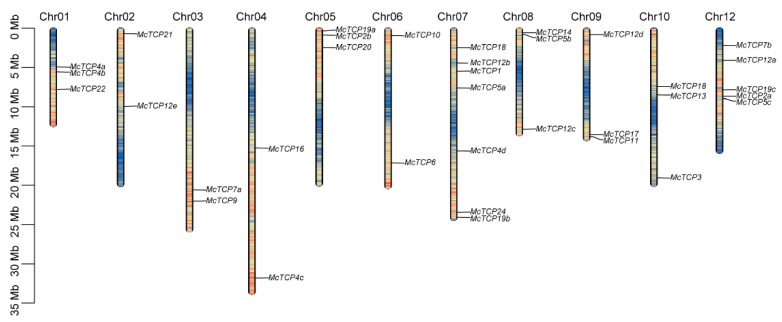
Gene location in different chromosomes. Length of the bars represents the size of the chromosome. Different color within each bar represents gene density on the chromosome. Red means high gene density, and blue means low gene density.

**Figure 2 molecules-27-09036-f002:**
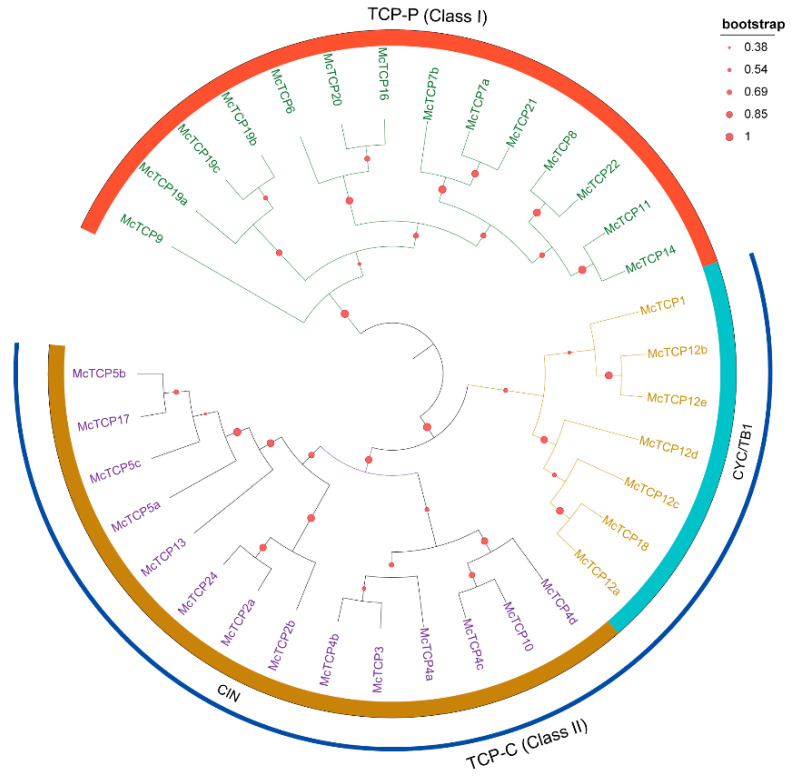
Phylogenetic evolution of *TCP* family members. Different colors of the outer ring denote the three main *TCP* genes. The blue arc represents *TCP-C*-type genes. The red dots on the tree branches represent bootstrap value. The size of the dots is in proportion to the bootstrap value.

**Figure 3 molecules-27-09036-f003:**
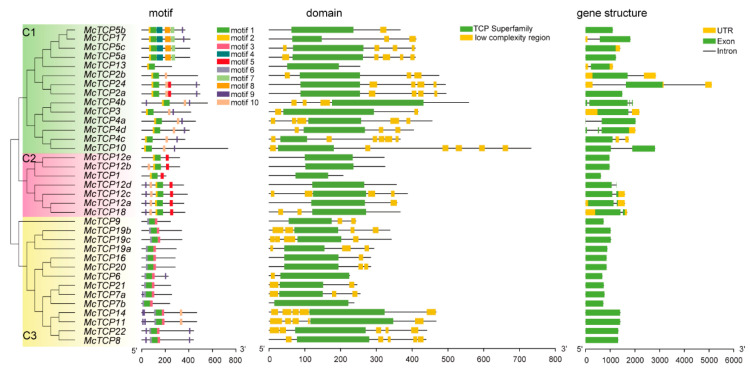
Motif, domain and gene structure analyses of TCP family members. Rectangles with different colors represent different motifs, domains, UTRs and exons, respectively.

**Figure 4 molecules-27-09036-f004:**
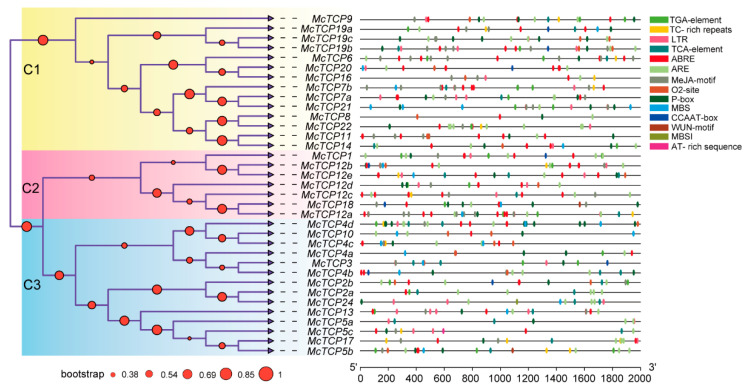
Promoter analysis of *TCP* family members. The red dot on branches of the phylogeny trees denotes bootstrap value. Different color models on the black lines mean elements of promoters.

**Figure 5 molecules-27-09036-f005:**
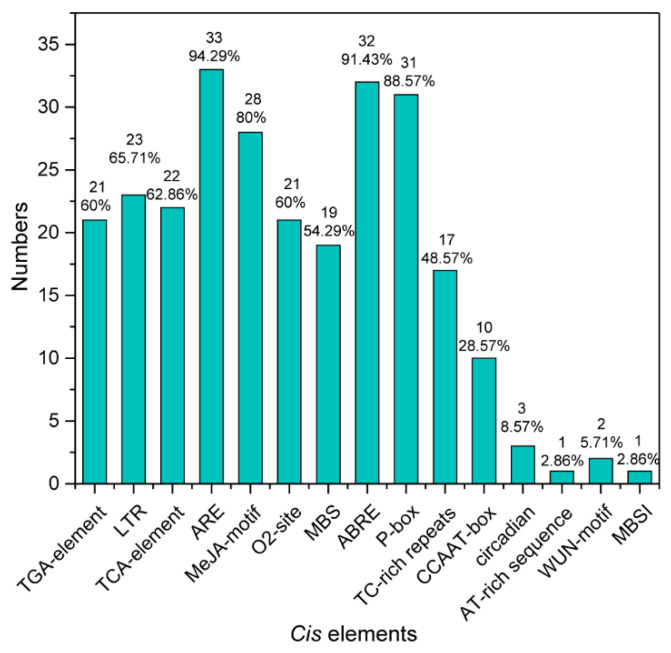
Statistics of different *Cis* elements in the promoter of the identified 35 *TCP* genes.

**Figure 6 molecules-27-09036-f006:**
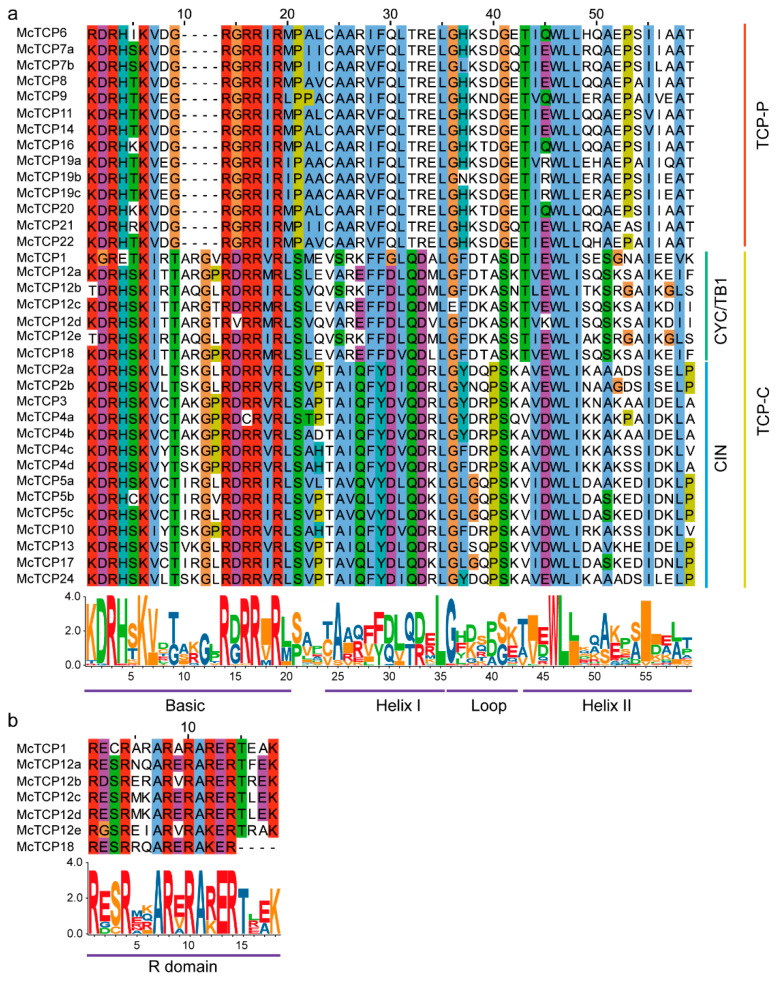
Sequence alignment and seqlogo of TCP TFs. (**a**) Alignment of the TCP superfamily domain in all identified TCP TFs; (**b**) Alignment of the R domain in all identified TCP TFs. The same color in the column and the big size of the letters in the seqlogo graph denote a high level of conservation in the corresponding position.

**Figure 7 molecules-27-09036-f007:**
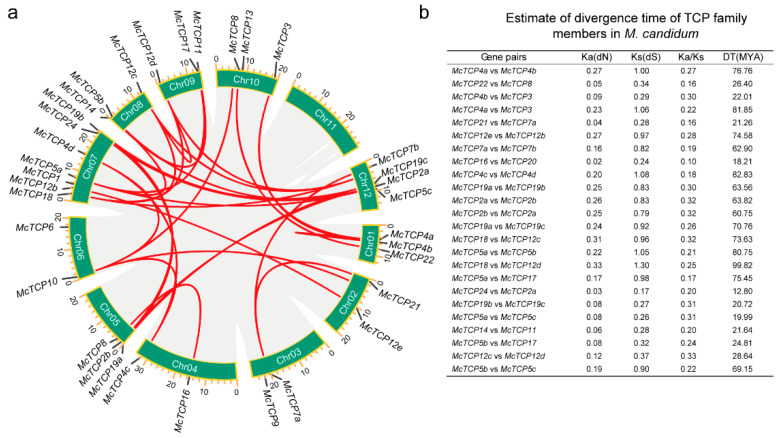
*TCP* gene pairs within *M. candidum*. (**a**) circos graph of *TCP* family members in *M*. *candidum*. Red lines within the circos graph represent a gene pair relationship. (**b**) Divergence time estimation of *TCP* gene pairs. Ka means non-synonymous substitution rate; Ks means synonymous substitution rate; DT means divergence time.

**Figure 8 molecules-27-09036-f008:**
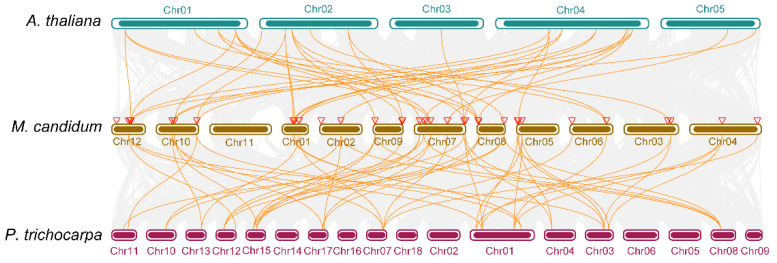
Multi-collinearity analysis of three species. The orange lines between two species denote collinearity genes among different species.

**Figure 9 molecules-27-09036-f009:**
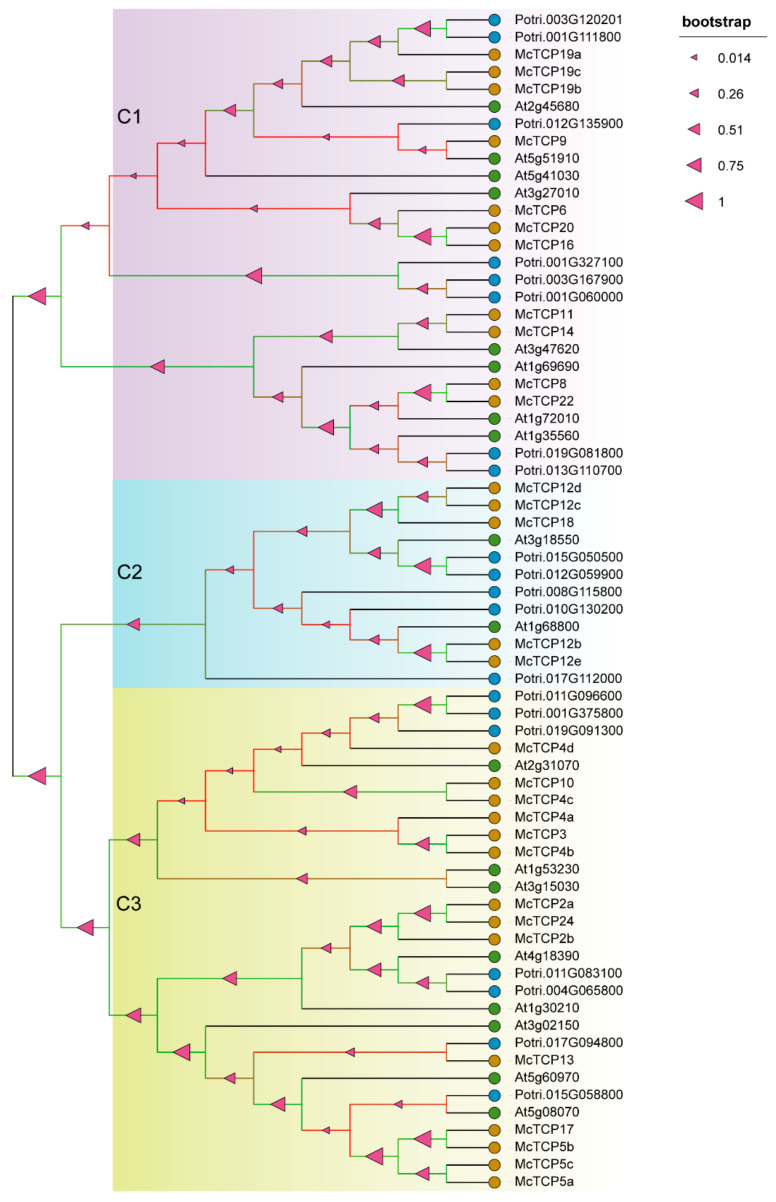
Phylogenetic evolution of *TCP* family members in three species. Different color models represent different clades. The pink triangle represents the bootstrap value, and the size of the triangle is in proportion to the bootstrap value. Before the gene name, blue dots represent *P*. *trichocarpa*, brown dots represent *M*. *candidum*, sapphire blue dots represent *A*. *thaliana*.

**Figure 10 molecules-27-09036-f010:**
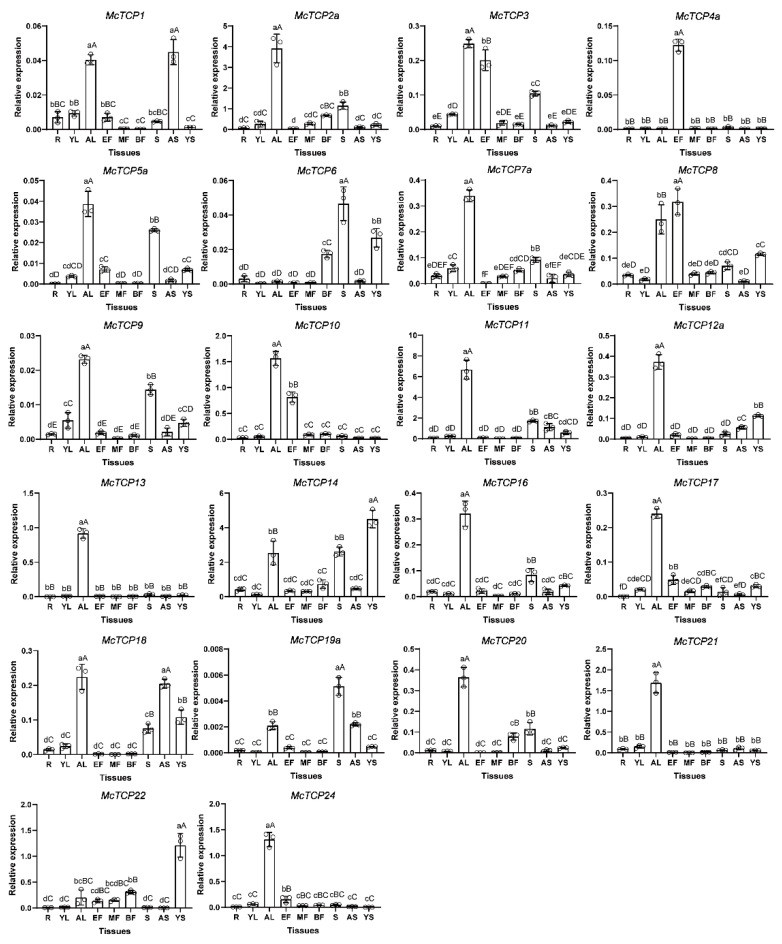
Expression analysis of *TCP*-family genes in nine tissues by qRT-PCR. YL: young leaves; AL: adult leaves; YS: young stems; AS: adult stems; S: seeds; R: roots; EF: early-stage flowers; MF: middle-stage flowers; BF: blooming flowers. Hollow circles in these graphs represent the relative expression of three biological repeats. The relative expression levels are shown as the means ± SDs. Duncan’s test was used to evaluate significant difference levels. Lowercase letters mean *p* ≤ 0.05, capital letters mean *p* ≤ 0.01.

## Data Availability

Not applicable.

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
