# Peer review of "Genome-Wide Identification and Characterization of TCP Gene Family Members in Melastoma candidum"

_molecules, 2022, doi:10.3390/molecules27249036_

Round 1

Reviewer 1 Report

In the manuscript entitled „Genome-wide identification and characterization of TCP gene 2 family members in Melastoma candidum” the authors showed the analysis of TCP gen family in Melastoma candidum. TCP proteins as transcription factors play a pivotal role during plant growth and development

In this manuscript authors present mainly bioinformatic analysis. The only experiment they present is a Q-PCR performed on RNA/cDNA for 22 from 35 TCP from different tissues. The presented formula for calculating the relative expression level is incorrect, no citation (line 473). What are the lengths of the obtained DNA fragments, there is no information about the PCR efficiency of each individual sample of primers.

Figure 6. Sequence alignment and seqlogo of TCP gene family. The same color in the column and 177 the big size of the letters mean a highly conservative region.” – no, this is not a DNA sequence alignment.

Authors should be more careful about are they writing about genes or proteins (italic usage) (fig. 3.; fig. 6; lines 67 and 70, and maybe some more).  

Author Response

In the manuscript entitled „Genome-wide identification and characterization of TCP gene 2 family members in Melastoma candidum” the authors showed the analysis of TCP gen family in Melastoma candidum. TCP proteins as transcription factors play a pivotal role during plant growth and development

Reply: We thank the reviewer for the review and the valuable comments. We have revised the manuscript based on your comments. Specific changes are listed below.

  1. In this manuscript authors present mainly bioinformatic analysis. The only experiment they present is a Q-PCR performed on RNA/cDNA for 22 from 35 TCP from different tissues. The presented formula for calculating the relative expression level is incorrect, no citation (line 473). What are the lengths of the obtained DNA fragments, there is no information about the PCR efficiency of each individual sample of primers.

Reply: 2ΔCt algorithm is also widely used in other research. Because our qRT-PCR was conducted within nine parallel tissues or developmental stages, there was no control group, so we think it is also reasonable to use this method. There are two papers were used as examples, which also used this algorithm. 1) Reverse breeding in Arabidopsis thaliana generates homozygous parental lines from a heterozygous plant (Wijnker et al 2012, Nature Genetics); 2) Transcriptional regulation of host NH4+ transporters and GS/GOGAT pathway in arbuscular mycorrhizal rice roots (Pérez-Tienda et al, 2014, Plant Physiol Biochem). A reference of methodology for calculating relative expression has been cited in the corresponding position.

According to the reviewer’s recommendation, PCR efficiency of each individual sample of primers should be performed. However, we think this process was mainly used to access primer quality, so we mixed the cDNA of nine tissues in equal amounts to calculate PCR efficiency following Pfaffl’s research (Relative expression software tool (REST©) for group-wise comparison and statistical analysis of relative expression results in real-time PCR). The obtained DNA length and primer PCR efficiency have been listed in Table S3.

  1. “Figure 6. Sequence alignment and seqlogo of TCPgene family. The same color in the column and 177 the big size of the letters mean a highly conservative region.” – no, this is not a DNA sequence alignment.

Reply: Yes. Sequence alignment and seqlogo were based on protein sequences and we have modified the title into “Sequence alignment and seqlogo of TCP TFs” and made some changes to the explanation part of the graph.

  1. Authors should be more careful about are they writing about genes or proteins (italic usage) (fig. 3.; fig. 6; lines 67 and 70, and maybe some more).  

Reply: We have corrected the gene name into italic in fig 3 and fig 4, line 67 and 70, however, in fig 2, 6, and 9 because we used the protein sequence to perform analysis, so here we think it is reasonable to use a regular font. And we also checked the whole paper for the font used. Thank you.

Reviewer 2 Report

The manuscript entitled “Genome-wide identification and characterization of TCP gene family members in Melastoma candidum” by Li et al. describes about the role of TCP gene family in different tissues of M. candidum. The results are interesting and could be useful for plant breeders yet, there are few comments which need to be tackled before its acceptance.

Abstract:

A clear objective of study is missing

Please indicate a clear take home message

Introduction

There no hypothesis in the concluding paragraph of introduction. I will suggest including a concrete hypothesis to indicate how the objective was achieved.

Results

Line 196-199, why results are being discussed in the results section? These can be moved to discussion section. Please check the whole results section and it should be separated from discussion part. Authors can take help from linking words or sentences to explain results.

There must be a significance level to show the expression of different gene in different tissue. Please perform ANOVA.

Materials and Methods

No information is available about statistical analysis. Like how ANOVA was performed and what packages were used to draw figures?

Author Response

The manuscript entitled “Genome-wide identification and characterization of TCP gene family members in Melastoma candidum” by Li et al. describes about the role of TCP gene family in different tissues of M. candidum. The results are interesting and could be useful for plant breeders yet, there are few comments which need to be tackled before its acceptance.

Reply: We thank the reviewer for the review and the valuable comments. We have revised the manuscript based on your comments. Specific changes are listed below.

  1. Abstract: A clear objective of study is missing, Please indicate a clear take home message

Reply: We have added a clear objective in the beginning part of the abstract. In addition, some qRT-PCR results were also added in the abstract.

  1. Introduction: There no hypothesis in the concluding paragraph of introduction. I will suggest including a concrete hypothesis to indicate how the objective was achieved.

Reply: We have added some concrete hypotheses in the concluding paragraph of the introduction.

  1. Results: Line 196-199, why results are being discussed in the results section? These can be moved to discussion section. Please check the whole results section and it should be separated from discussion part. Authors can take help from linking words or sentences to explain results.

Reply: We have moved the Line196-199 to discussion part and checked the whole result sections.

  1. There must be a significance level to show the expression of different gene in different tissue. Please perform ANOVA.

Reply: We have performed ANOVA analysis and added letters in Figure 10 to show significant difference levels. Thank you.

  1. Materials and Methods

No information is available about statistical analysis. Like how ANOVA was performed and what packages were used to draw figures?

Reply: We have added statistical analysis part in 4.7. Because most of the analyses in this paper were based on TBtool software, which is a comprehensive software for bioinformatic analysis and visualization, there were no other packages used. In the materials and methods part, we have mentioned about analysis process and visualization, For example, in the following paragraphs

4.2. Phylogeny tree construction and location of TCP gene family members in chromosome

All the identified TCP genes were aligned by MUSCLE method in MEGA 6.01, and then a phylogenetic tree was constructed by the ML (maximum likelihood) method based on LG models with 1000 bootstrap replications. The phylogeny tree was visualized on the iTOL website (https://itol.embl.de/). By utilizing the Gene Location Visualize from GTF/GFF function module of TBtools software, we visualized the distribution of TCP genes along the chromosomes through the gtf annotation of the genome and the gene density file.

4.3. Visualization of motif, domain, gene structure, and promoter of TCP genes

All the identified TCP genes were conducted motif, domain, gene structure, and promoter analyses. The upstream 2000 bp of the TCP CDs were extracted for the purpose of conducting promoter analyses on the website (https://bioinformatics.psb.ugent.be/webtools/plantcare/ html/). With the gff file of the genome, gene structures including CDs and untranslational region (UTR) were displayed in the Gene structure view of TBtools. Based on the Identification of TCP transcription factors part, the motifs, and domains of TCP genes were visualized by the Gene structure view and Batch SMART module of TBtools software, respectively.

Round 2

Reviewer 2 Report

The manuscript entitled “Genome-wide identification and characterization of TCP gene family members in Melastoma candidum” by Li et al. describes about the role of TCP gene family in different tissues of M. candidum. Many thanks for incorporating the suggested changes yet, a clear hypothesis is missing in the current study. Again, I will suggest including a concrete hypothesis in the concluding paragraph of introduction to indicate how the objective was achieved.

Author Response

Thanks a bunch for your patience and timely response. We added one sentence at the beginning of the introduction's concluding paragraph: "Although TCP genes have important regulatory roles in plant growth and development, abiotic stress responses, and hormone metabolism, limited information is available on M. candidum." Other changes were also marked with track changes.  Thanks again!